# The Impact of Maternal Prenatal Stress Related to the COVID-19 Pandemic during the First 1000 Days: A Historical Perspective

**DOI:** 10.3390/ijerph19084710

**Published:** 2022-04-13

**Authors:** Sam Schoenmakers, E. J. (Joanne) Verweij, Roseriet Beijers, Hilmar H. Bijma, Jasper V. Been, Régine P. M. Steegers-Theunissen, Marion P. G. Koopmans, Irwin K. M. Reiss, Eric A. P. Steegers

**Affiliations:** 1Department of Obstetrics and Gynaecology, Division of Obstetrics and Fetal Medicine, Erasmus MC, University Medical Center Rotterdam, 3015 GD Rotterdam, The Netherlands; e.j.t.verweij@lumc.nl (E.J.V.); h.bijma@erasmusmc.nl (H.H.B.); j.been@erasmusmc.nl (J.V.B.); r.steegers@erasmusmc.nl (R.P.M.S.-T.); e.a.p.steegers@erasmusmc.nl (E.A.P.S.); 2Department of Obstetrics, Leiden University Medical Center, 2333 ZA Leiden, The Netherlands; 3Social Development, Behavioural Science Institute, Radboud University, 6525 XZ Nijmegen, The Netherlands; roseriet.beijers@ru.nl; 4Cognitive Neuroscience, Donders Institute, Radboud University Medical Center, 6525 GA Nijmegen, The Netherlands; 5Department of Paediatrics, Division of Neonatology, Erasmus MC Sophia Children’s Hospital, University Medical Center Rotterdam, 3015 GD Rotterdam, The Netherlands; i.reiss@erasmusmc.nl; 6Department of Public Health, Erasmus MC, University Medical Center Rotterdam, 3015 GD Rotterdam, The Netherlands; 7Department of Viroscience, Erasmus MC, University Medical Center Rotterdam, 3015 GD Rotterdam, The Netherlands; m.koopmans@erasmusmc.nl

**Keywords:** first 1000 days, COVID-19, prenatal stress, hypothalamic–pituitary–adrenal axis, mitigation measures, DOHAD, pregnancy, child outcomes

## Abstract

The COVID-19 pandemic has a major impact on society, particularly affecting its vulnerable members, including pregnant women and their unborn children. Pregnant mothers reported fear of infection, fear of vertical transmission, fear of poor birth and child outcomes, social isolation, uncertainty about their partner’s presence during medical appointments and delivery, increased domestic abuse, and other collateral damage, including vaccine hesitancy. Accordingly, pregnant women’s known vulnerability for mental health problems has become a concern during the COVID-19 pandemic, also because of the known effects of prenatal stress for the unborn child. The current narrative review provides a historical overview of transgenerational effects of exposure to disasters during pregnancy, and the role of maternal prenatal stress. We place these effects into the perspective of the COVID-19 pandemic. Hereby, we aim to draw attention to the psychological impact of the COVID-19 pandemic on women of reproductive age (15–49 year) and its potential associated short-term and long-term consequences for the health of children who are conceived, carried, and born during this pandemic. Timely detection and intervention during the first 1000 days is essential to reduce the burden of transgenerational effects of the COVID-19 pandemic.

## 1. Introduction

The COVID-19 pandemic has a major impact on society, particularly affecting its vulnerable members, including pregnant women and their unborn children. Pregnancy involves profound physiological and psychological changes in women. As such, pregnancy is known to be a sensitive period for developing symptoms of anxiety, depression, and stress [1]. Before COVID-19, considerable numbers of pregnant women, ranging between 7 to 40% [2], suffered from mental health problems [2,3]. These numbers increased considerably during the COVID-19 pandemic [4,5,6]. Besides reports of reduced numbers of preterm birth [7] and positive effects of working from home, pregnant women seemed to psychologically suffer from the pandemic, as they reported fear of infection, fear of vertical transmission, fear of adverse birth and child outcomes, social isolation, uncertainty about their partner’s presence during medical appointments and delivery, increased domestic abuse, and other collateral damage, including vaccine hesitancy [4,5]. It is thus not surprising that, in The Netherlands, for example, a twofold increase was reported in the prevalence rate of women experiencing mental health symptoms during the first lockdown. More than 1 in 10 women experienced clinically relevant symptoms of depression, and nearly 1 in 2 women experienced clinically relevant symptoms of anxiety [6]. Accordingly, pregnant women’s known vulnerability for mental health problems has become a concern during the COVID-19 pandemic, also because of the known effects of prenatal stress for the unborn child. The current narrative review illustrates the transgenerational effects caused by disasters in order to highlight that special attention should be paid to potential, but less obvious, consequences of the COVID-19 pandemic, namely those consequences on children conceived, carried, and born during the pandemic.

The first wave of COVID-19 in early 2020 led to the publication of daily-to-weekly revised national and international COVID-19 and pregnancy guidelines based on either historical or very limited current insight and knowledge. Initial recommendations were aimed at reducing transmission and infection risks, fueled by data from previous corona virus (SARS-CoV and MERS-CoV) outbreaks that showed impressive mortality rates, up to 20%, in infected pregnant women [8,9]. Several professional associations in different countries, including The Royal College of Obstetrics and Gynaecology (UK), the American College of Obstetrics and Gynaecology (USA), and the Dutch Society of Obstetrics and Gynaecology (The Netherlands), quickly responded with recommendations for expecting parents, such as minimizing hospital visits and restricting outpatient clinic consultations, remaining indoors and self-isolating when symptomatic for at least 7 days, delaying routine obstetrical care until after the self-isolation period, physical distancing between physician and patient, restricting hospital deliveries, and restrictions around the presence of partners and significant others during labor [10,11,12]. Next to unprecedented changes in obstetrical care, it soon became apparent that pregnant women with COVID-19 disease, as compared to non-pregnant women of a similar age, were at increased risk for admission to the intensive care unit and mechanical ventilation [13]. Moreover, SARS-CoV-2 infection during pregnancy can lead to placental infection, resulting in massive placental necrosis [14], which is related to fetal distress and an increased risk for stillbirth [15,16]. On top of all that, the availability of the COVID-19 vaccines placed additional psychological burden on pregnant women, due to vaccine hesitancy [17,18,19]. Vaccine uptake during pregnancy is, in general, confronted with a variety of barriers [20]. Due to the exclusion of pregnant women from initial vaccination clinical trails [21], the evidence of safety and efficacy of COVID-19 vaccination during pregnancy was lacking. Therefore, with the introduction of the COVID-19 vaccines, many countries were not offering COVID-19 vaccines to pregnant women [22]; however, this changed after the Centers for Disease Control and Prevention report in December 2020 [23,24]. All of these consequences of the COVID-19 pandemic and its mitigation measures for pregnant women likely contributed to increased levels of psychological stress.

It is recognized that maternal stress during pregnancy is associated with offspring (neuro)developmental outcomes during the life course, such as socioemotional problems, autism spectrum disorder (ASD), and attention deficit hyperactivity disorder (ADHD) [1,25,26]. This phenomenon is also described in the Developmental Origins of Health and Disease (DOHaD) paradigm [27] introduced by Barker and colleagues in 1986 [28]. The DOHaD paradigm states that sensitive windows during fetal development exist, in which tissue development and function in utero and during the early postnatal life can be modified by external environmental exposures, resulting in increased risks for adverse (neuro)developmental and health outcomes [29,30]. This latter process is also often referred to as fetal programming, because many effects are profound, long-term, and seem to be transmitted from the offspring to the next generation [1,31].

While research has come a long way in describing relations between maternal prenatal stress and offspring outcomes, much less is known about the mechanisms through which maternal prenatal stress would exert programming effects on the fetus. However, a role is proposed for the maternal hypothalamic–pituitary–adrenal (HPA) axis, inflammation, and gut microbiome and placental dysfunction [31] (Figure 1). These mechanisms are discussed in greater detail in the Discussion section. Moreover, the transgenerational health and socioeconomic effects of in utero exposure to increased levels of prenatal maternal stress due to previous well-documented disasters, such as the Spanish Flu pandemic and Queensland flooding, are still evident [32,33,34,35,36,37]. We believe that the transgenerational lessons learned from historical experiences with disasters should quickly be applied to the current COVID-19 pandemic in order to prevent or timely detect similar long-term consequences. The COVID-19 pandemic meets the United Nations 2009 definition of disaster, which is defined as a serious disruption of the functioning of a community or society, causing widespread human, material, economic, or environmental losses which exceed the ability of the affected community or society to cope by using its own resources [38].

In general, disasters can be classified into two types, and they either result from natural causes, i.e., natural disasters, or from human actions, i.e., manmade disasters [39], subdivided into diverse subtypes (Table 1). We aim to offer a historical perspective by presenting a narrative review of known effects of prenatal maternal stress on the unborn generation caused by previous disasters, both natural and manmade, which occurred in the 20th and 21st centuries. Hereby, we want to draw attention to the psychological impact of the COVID-19 pandemic on the reproductive population and its potential associated short-term and long-term consequences during the first 1000 days of life: the health of children who are conceived, carried, and born during the current global crisis. Ultimately, we envision that this narrative review will initiate timely attention to and registration of the consequences of the pandemic for the COVID-19 generation. This approach will increase our understanding of the effects of maternal prenatal stress and put secondary prevention and intervention programs in place during these precious first 1000 days, which form the foundation for the remaining life course, and beyond. The latter is especially relevant, as impaired child neurodevelopment and associated outcomes due to prenatal stress can be improved by high-quality maternal caregiving [40,41] and improved healthcare [42,43].

## 2. Materials and Methods

Searches were performed in the databases of Medline, Embase, PubMed, Web of Science, and Google Scholar between January 1900 and December 2021. The search-strategy terms used included, but were not limited to, “prenatal stress”, “preconception period”, “transgenerational effects”, “natural disaster”, “manmade disaster”, “disaster”, “trauma”, and “pandemic”. These were combined by using the Boolean operator “or”. As it is a narrative review, the inclusion and exclusion criteria were not strictly set. However, only original articles in the English language were included. To present an overview of effects of exposure to a disaster in general, a distinction was made into natural and manmade disasters (Table 1). As each disaster can differ between moment of occurrence, cause, intensity, extent, or duration of exposure, a selection of the literature search was made based on the review of titles and abstracts. A selection of published literature on the effects of disaster, either natural or manmade, was made by SS and EJV based on (1) the time period of occurrence (i.e., aiming at different periods of the 20th and 21st century), (2) the inclusion of examples of several subtypes of natural disasters and manmade disasters, (3) studies including follow-up or time between disaster and investigation of at least 5 years, and (4) the inclusion of well-known disasters to appeal to imagination and recognition.

## 3. Results

### 3.1. Natural Disasters

#### 3.1.1. Biological

##### The 1918 Influenza Pandemic (1918/1919)

The 1918 influenza pandemic, also known as the Spanish Flu, started at the beginning of 1918 and ended after four waves at the beginning of 1920.

In the USA, the 1918 pandemic influenza was highly concentrated from October 1918 until January 1919, and offspring potentially exposed in utero to maternal influenza infection were mostly born between January and September 2019. Several cohorts in the USA of offspring with potential in utero exposure to the 1918 influenza were compared to offspring that were in utero beyond the height of the 1918 pandemic. The results indicate significant adverse effects on several health and socioeconomic parameters across the life course for in utero exposure to 1918 influenza. Studies performed in the United States showed increased rates of disability [44], functional limitations in hearing, speaking and walking, and more diabetes and cardiovascular disease [45]. All were accompanied by lower educational attainment, income, and socioeconomic status [44]. A similar analysis that combined several cohorts of prenatal exposure to the pandemic in Taiwan indicated that the offspring were shorter in length, as children and as adolescents, more prone to diabetes, respiratory, circulatory, and kidney problems, especially later in life, and were less educated [34]. A study performed in a comparable cohort in Sweden showed comparable results [33]. In addition, they showed that the effects were skewed depending on sex and gestational age of exposure: adverse health comes were limited to males exposed to the pandemic during the second trimester and to females exposed during late pregnancy until the first months after birth [33]. Recently, Cock and colleagues explored transgenerational effects of prenatal exposure to the 1918 influenza pandemic and showed large effects on both health and economic outcomes over multiple generations [32]. In sum, in utero exposure to the 1918 influenza pandemic, especially during mid- and late gestation, was associated with multiple morbidities across the lifespan and lower socioeconomic status, and these associations seem to spill over to the next generation(s) [32,33,34,35].

#### 3.1.2. Geological

##### Tangshan Earthquake, China (1976)

On 28 July 1976, the city of Tangshan (China) was hit early in the morning by an earthquake with a magnitude of 7.8 on the Richter scale. Around 250,000 people were killed, with hundreds of thousands of people injured and massive structural damage. Several studies have been performed into the effects of prenatal exposure to the Tangshan earthquake, which include an association with higher serum uric acid and increased risk for hyperuricemia in early adulthood. Elevated levels of serum uric acid appears to be linked to an increased risk of end-stage renal failure, hypertension, cardiovascular disease, and diabetes mellitus [46,47,48,49,50,51,52]. Exposure in utero during the first trimester seems to increase the risk for diabetes and schizophrenia in adulthood, while fetuses exposed during the second and third trimester showed increased risk for impaired visual spatial memory, reduced working memory, and increased risk for cardiovascular disease in adulthood [53,54,55,56,57,58].

#### 3.1.3. Hydrometerological

##### Quebec Ice Storm (1998), Hurricane Katrina, Louisiana (2005), Superstorm Sandy (2012)

In January 1998, the Great Ice Storm of 1998 struck the eastern parts of Canada, including Quebec, and the United States, causing massive damage to the electrical infrastructure resulting in power outages ranging from days to weeks. Research into long-term offspring effects of prenatal exposure to the Ice Storm of Quebec has shown diverse effects on postpartum development, including poorer cognitive and language development, independent of maternal personality factors [59], higher body mass index, and central adiposity at different ages [60], suggested to be mediated by different DNA methylation [60,61], and long-term dysregulation of the hypothalamic–pituitary–adrenal (HPA) axis [62].

Likewise, in August 2005, class 5 Hurricane Katrina made landfall and caused catastrophic damage, especially in the city of New Orleans, leaving thousands of non-evacuated people bereaved of food and shelter. More recently, in 2012, Superstorm Sandy caused extensive damage in the Caribbean, Bahamas, United States, and Canada, with thousands of people having no access to electricity, housing, and food. Investigation into Hurricane Katrina–related maternal stress has found no large associations between maternal stress due to this natural disaster and infant temperament. However, maternal mental health problems were associated with reports of difficult temperament in their offspring [63]. On the other hand, Hurricane Katrina exposure was significantly associated with induction of labor and perceived maternal stress [64], of which the latter is known to significantly predispose to pregnancy-induced hypertension and gestational diabetes [65]. Superstorm Sandy seemed to magnify prenatal maternal depression, and the interaction between exposure to Sandy and maternal depression was associated with lower emotional regulation and greater distress at six months in neonates [66,67].

#### 3.1.4. Floods

##### Iowa Flood (2008) and The QF2011 Queensland Flood (Brisbane) (2011)

In 2008, Iowa, the Midwest of the US, flooded from the beginning of June till July 2008, with many people losing their homes and experiencing a lack of public and social services, while polluted water and remaining debris after the receding waters caused a lot of sanitary problems. At the end of 2010, Queensland, Australia, was hit by a series of floods, resulting in mass evacuation, affecting over 200,000 people. The offspring of pregnant women exposed to severe flooding have shown greater BMI increase; higher risk for central adiposity [68,69]; associations with lower cognitive abilities in 30-month-old toddlers, but better language abilities in 30-month-old boys [70]; and long-term alteration in the functioning of the HPA axis, including higher cortisol awakening responses, higher total daily cortisol secretion, and higher stress reactivity, evident from early ages onward [36,71,72]. Moreover, greater pregnant women’s flood-related objective and subjective stress predicted worse and delayed developmental outcomes for fine motor and problem-solving skills at six months of infant age [43]. In addition, based on questionnaires to assess maternal stress level, higher reported maternal stress was associated with an altered placental glucocorticoid system, as measured by a reduced presence of glucocorticoid receptor mRNA, which can be linked to reduced fetal protection against prenatal maternal cortisol [36].

#### 3.1.5. Manmade Disasters

##### Famines

The Dutch Famine, (“Dutch Hunger Winter”) (1944/1945); Great Leap Forward Famine, China (1959–1961); and Biafra Famine, Nigeria (1967–1970): During the last century, several manmade famines have occurred. In 1944/1945 the Dutch famine was the result of a blockage of food by the German army, causing starvation. Around 4.5 million people were affected. In the period 1959–1961, the policies of a new economic and social campaign, the Great Leap Forward, led to widespread famine in China, with millions of casualties. During the Nigerian civil war, the transport of food to Biafra was blocked from 1967 until 1970, resulting in the starvation of millions of people.

The many results of the Dutch Hunger Winter Study have shown that fetal exposure to malnutrition has effects across the lifespan [73]. The lifelong effects on health were related to the timing of exposure during gestation (i.e., first, second, or third trimester), as well as the course of the catch-up period [74]. It appeared that exposure to the Dutch Hunger Winter during the first trimester was associated with the poorest outcomes, including coronary heart disease, raised serum lipids, obesity, and breast cancer. Exposure to malnutrition during the first trimester also increased the risk for schizophrenia for female offspring during their postpartum life course [75,76,77]. Mid-gestation exposure was associated with reduced birth weight, obstructive airways disease, and microalbuminuria during the life course [78]. The fetuses exposed during late gestation were born with reduced birth weight and remained smaller throughout life [73]. Results of the less investigated Chinese famine indicated that fetal exposure to famine was associated with an increased risk for hypertension [79], type 2 diabetes mellitus [80], and schizophrenia later in life [81]. Effects of famine exposure in early life on body mass index (BMI) and blood pressure were also found to persist in the next generation(s) [82]. Moreover, exposure to food deprivation during the fetal and infant period in Biafra, a former secessionist state in West Africa, was associated with an increased risk of hypertension and impaired glucose tolerance at the age of 40 years [83].

##### The 9/11 Terrorist Attacks (2001)

In 2001, terrorists hijacked four airplanes and attacked several targets in the United States on 11 September, changing the lives of millions forever. Offspring of women who were pregnant and present near or at the World Trade Center (WTC) and who developed posttraumatic stress disorder after the WTC attacks showed lower levels of salivary cortisol in the first year of age. The effect was most prominent in neonates who were exposed to the WTC attacks in the third trimester [84].

##### Tailings Dam Breakage, Brazil (2015)

In 2015, a tailings dam in Marina (Brazil) failed, resulting in a massive flooding that impacted millions of lives, referred to as the Marina Tragedy. It was found that pregnant women directly exposed to the dam breakage experienced a shortened length of gestation and an increase in preterm birth. Due to the economic depression following the disaster, it was not possible to deduce if these results were caused by the exposure to the Marina Tragedy itself or its aftermath [85].

## 4. Discussion

Regardless of its cause or length, exposure to a disaster during pregnancy is associated with increased risk of adverse outcomes for the offspring (see Table 2). While some disasters result from natural causes, i.e., natural disasters, others are the result of human actions, i.e., manmade disasters [39]. Likewise, while some disasters involve events with a high impact during a short period of time, such as a flooding, storm, or earthquake, other disasters comprise longer periods of time, such as a pandemic, war, or famine. However, as can be concluded from this narrative review, all disasters, independent of their cause, can have short- and long-term impacts on the next generation(s). The current COVID-19 pandemic should be viewed as a combination between a natural disaster, initiated by the global spreading of the SARS-CoV-2 virus and its diverse variants, and a manmade disaster, resulting from the collateral damage caused by national COVID-19 mitigation measures. The COVID-19 pandemic also combines a short period with sudden, high impact (i.e., the first wave of COVID-19 and its related lockdown measures in spring 2020), with a long period of impact (e.g., the different waves of COVID-19, its different mitigation measures and the (in)direct consequences, and the overcrowding and delaying of health care), with long-term still-unclear, overarching effects for global society and economy. As a result, it is likely that also the COVID-19 pandemic is associated with negative consequences for the health of children who are conceived, carried, and born during this global crisis. Below, we discuss the potential risks for our future generation(s), due to maternal exposure to the COVID-19 pandemic during pregnancy, including the role of maternal stress and its underlying mechanisms.

### 4.1. Prenatal Maternal Stress and Developmental Origins of Health and Disease (DOHaD)

The DOHaD paradigm [27] states that prenatal maternal stress can negatively affect pregnancy outcomes and fetal developmental processes, including neurogenesis, neural migration, and myelination [86]. Adverse outcome of these fetal neurodevelopmental processes are associated with an increased risk of developing mental and physiological disorders later in life, such as autism spectrum disorder and attention-deficit/hyperactivity disorder [26,87]. For example, maternal prenatal stress associated with the potential prospect of deportation in pregnant Latina women during the 2016 USA presidential election appeared to be associated with an increase in the incidence of preterm birth among these women [88], and the offspring of pregnant women who developed posttraumatic stress disorder (PTSD) due to the terrorist attacks on the World Trade Center showed lower levels of cortisol when compared to the offspring of mothers who did not. Since reduced levels of cortisol are linked to susceptibility to PTSD [89], these offspring are potentially at an increased risk of developing PTSD when exposed to traumatic events themselves later in life [84].

The maternal hypothalamic–pituitary–adrenal (HPA) axis and its most studied end product cortisol have, for a long time, been proposed as the most important mechanism responsible for the effects of maternal prenatal psychological stress on offspring outcomes [31]. Prenatal maternal stress activates the maternal HPA axis, resulting in the increased production and circulation of cortisol, with direct and indirect effects on the developing fetus and its brain (Figure 1), as maternal cortisol is known to be able to pass the placental barrier [31,90,91,92]. However, several other studies did not show a significant association—or showed only a weak association—between prenatal maternal cortisol levels and offspring developmental outcome [93], while maternal prenatal psychosocial stress often independently predicts infant outcomes irrespective of maternal biomarkers of HPA axis functioning, such as cortisol reactivity or the cortisol output during the day [31].

These findings suggest that additional mechanisms related to prenatal maternal stress, besides cortisol physiology, are affecting fetal neurodevelopment (Figure 1). One of the mechanisms that have been proposed is the maternal gut microbiome [94], since an association seems to exist between maternal anxiety and the composition of the maternal gut microbiota during pregnancy [95]. The microbial composition of the maternal gut will directly affect the maternal prenatal metabolism, which is essential for fetal growth and development. Importantly, the same maternal gut microbes will be horizontally transmissed during a vaginal delivery. Neonatal gut colonization occurs initially via transmission of the maternal gut and skin microbiome and is essential for the latter part of the first 1000 days, since the development and maturation of the neonatal gut microbiome and neonatal cognitive development occur in parallel, due to the microbial control of neuromodulatory metabolites production, such as GABA and serotonin [96,97]. The effects of the maternal microbial neuromodulatory components on fetal neurodevelopment remain, up until now, unknown. Other potential prenatal stress mechanisms related to (neuro)development during the first 1000 days include compromised placental function (e.g., an inability to upregulate 11b-HSD2 expression necessary to reduce fetal exposure to cortisol [98,99,100]; increased catecholamines [101]; dysfunctional maternal immune system functioning; and maternal lifestyle and nutritional behaviors, including poor sleep and lack of exercise. Finally, epigenetics and the postnatal setting, including nourishment, caregiving, and other environmental influences, could potentially mediate the association between prenatal maternal stress and compromised fetal and offspring development [31,102]. In summary, prenatal maternal stress has been associated with compromised fetal (neuro)development and, with it, off-spring behavioral, physiological, and neurological development during the individual’s first 1000 days and beyond (Figure 1).

### 4.2. COVID-19 Pandemic, Maternal Prenatal Stress, and the First 1000 Days

Prenatal exposure to negative maternal psychological factors, including stress due to disasters, is thus associated with a range of poor outcomes for the offspring, such as preterm birth, risk for schizophrenia later in life, and a dysbiotic gut microbiome, with some effects even lasting until the end of the life course [86,88,94,96,97,103,104,105,106,107,108,109,110,111,112,113,114,115,116,117,118,119,120,121,122,123,124]. However, it is important to mention that disasters often involve a combination between psychological stressors and physical strain, such as during a famine. The scarce resources of food during a famine will indispensably cause psychological stress due to uncertainty about the availability of food, the hopelessness of the period, the feeling of hunger, and the physical weakening by the continued lack of nutrients. However, it is difficult to distinguish whether the poor outcomes seen in the offspring are the result of maternal prenatal stress and/or the physical strain, such as the famine and persistent cold in the case of the Dutch Hunger Winter.

Disentangling prenatal stress from physical strain also applies to COVID-19-related prenatal stress research. Clinical reports have shown that COVID-19 infection does seem to affect pregnant women more severely than age-matched non-pregnant patients [13,105,112], while vertical transmission seems to be very limited [113]. COVID-19 infection during pregnancy is also associated with adverse obstetrical outcomes, such as preeclampsia, (iatrogenic) preterm birth, and low birth-weight [125,126,127]. In addition, pregnant women with COVID-19 are at risk for placental dysfunction, characterized by inflammatory changes, massive perivillous fibrin depositions and placental necrosis, and related fetal distress, specifically when infected with the Delta variant [14,15,16]. It is well-known that maternal infection, in general, during pregnancy poses a threat for the development of a range of mental disorders in the offspring, such as schizophrenia, autism, and attention deficit/hyperactivity disorder [104,106,109,111,114,120]. It has been suggested that maternal inflammation caused by an intruding pathogen can interfere with fetal brain development by increased levels of interleukin-6 [116,118,128]. Severe cases of COVID-19 infection are indeed associated with high levels of interleukin-6 [121]. As such, mechanistic understanding of maternal COVID-19-related prenatal stress and COVID-19 infections during pregnancy overlap. Trying to disentangle the different mechanisms will be challenging, but should be an important goal for future research.

Given the mostly unexpected and unannounced nature of a disaster, other challenges are also involved in disaster research. Most previous disaster studies were not able to obtain measurements of stress-coping mechanisms and disaster-related stress in, respectively, the days and weeks before and directly following the event. If measures of maternal stress at a later time were included, it probably can be assumed that individuals reporting high stress would have also experienced high stress at the time of the event. However, the opposite does not necessarily uphold: women who have reported no or low levels at a later time point might still have experienced high levels of stress at the time of the event. After the announcement of the first lockdown, many prenatal-stress researchers quickly developed and launched online questionnaires to investigate maternal objective and subjective stress and other COVID-19-related changes (e.g., changes in healthcare, work, and lifestyle), such as the Pregnancy during the COVID-19 Pandemic (PdP) study [129] and the COVID-19 and Perinatal Experience (COPE) study [6]. The results of these, as well as other similar studies, will prove to be valuable to analyze independent effects of objective stressors, as well as maternal subjective stress from other COVID-19-related changes, such as working conditions. The outcomes can also be used for association studies, during the first 1000 days and beyond, to study short- and long-term consequences for the offspring.

**Table 2 ijerph-19-04710-t002:** An overview of the effects of prenatal maternal stress during the first 1000 days, due to exposure to disasters.

Disaster	Duration Disaster	Type	Offspring Effects during the Life Course	Sample Size	Outcome Variables	Confounders/Limitations
1918 Influenza Pandemic (1918/1919)	1–2 years	Natural	Exposure during mid- and late gestation is associated with multiple morbidities later in life, such as depression, diabetes, renal disease, ischemic heart disease, higher mortality, and lower socioeconomic status (SES) [32,33,34,35].	I. Wisconsin Longitudinal Survey (1957, second-generation USA) [32] II. Swedish population database (1968–2012) [33] III. Taiwan population cohort (1916–1926) *n* = 870.468 [34] IV. National Health Interview Surveys (NHIS, USA) *n* = 101,068 [35]	I. First generation: flu indicators (male/female/either sex), controls (year of birth, male/female), dependent variables (male/female years of schooling, job prestige, and family SES). Second generation: controls (female indicator, birth year and order), dependent variables (years of schooling, family income, net worth, self-reported health, height, and BMI). Third generation: controls (birth year, female indicator, birth order, dependent variable (years of schooling). II. Occupational status 1970 (men), income (men), hospitalization (duration; men/women), mortality cause (men/women), birth year, and exposure trimester 1–3. III. Anthropometric outcome elementary school/middle school, educational attainment and outcome, health attainment and outcome, height in childhood and adolescents, and maternal mortality IV. Health outcome (medically important condition, e.g., diabetes and cardiovascular disease).	I. The 1918 pandemic co-occurred with a world war. II. Not fully able to distinguish between socioeconomic and biological factors. III. Possible positive selection due to high infant mortality rates (18%). IV. Underestimated effect expected higher mortality bore age 60, self-reported data.
The Dutch Famine, (“Dutch Hunger Winter”) (1944/1945)	±5 months	Natural and manmade	Life-long effects on health; effects depending on the timing of exposure in gestation and the course of the catch-up period [74]. Exposure during early gestation is related to coronary heart disease, raised serum lipids, and more obesity, whereas mid-gestation exposure is associated with obstructive airways disease and microalbuminuria during the life course [78]. Twice more likely to develop schizophrenia [75,76,77].	I. Review of the literature [74] II. *n* = 100,543 [78] III. *n* > 40.000 [75] IV. *n* > 40.000 [76] V. *n*= 2414 [77]		I. Review of the literature. II. Ecological measurement of nutritional status and date of conception not measured, and preterm labor is risk factor, too. III. Selective conception, selective survival. IV. The ascertainment of the population at risk, the ascertainment of cases [2], the reliability of diagnosis [3], the degree of food deprivation [4], the control of coincident factors [5], the use of group vs. individual data [6], and the control of social class [7]. V. Pinpointing the exact timing of famine exposure during gestation and associated outcomes in later life, due to the relatively small sample size and partial overlap between the three famine exposed groups on the other.
Great Leap Forward Famine, China (1959–1961)	±3 years	Natural and manmade	Prenatal exposure increases risk for schizophrenia later in life [81].	1987 Chinese National Disability Sample Survey *n* = 1,579,316	Chinese National Disability Sample Survey (CNDSS), with focus on disability. Two step: interview + questionnaire. On-site diagnosis professional psychiatrists.	Large-scale sample survey, accuracy on diagnosis might not as a psychiatric clinic, selected cohorts were adults 22–32 years, and researchers not able to look at cohort differences in schizophrenia rate beyond this age.
Biafra Famine, Nigeria (1967–1970)	±3 years	Manmade	Exposure during fetal and infant period is associated with increased risk of hypertension and glucose tolerance [83].	*n* = 1339	Observational study, men and non-pregnant women. Level of education, current smoking, previously diagnosed hypertension, diabetes, treatment. Level of education. Blood pressure and random plasma glucose. Height, weight, BMI, and waist circumference.	The lack of anthropometric data at birth and in infancy.
Tangshan earthquake, China (1976)		Natural	Higher serum uric-acid concentrations; 70% more likely to develop hyperuricemia approximately 33 years after the event, independent of traditional risk factors. [130]	*n* = 536	Collection of blood samples for uric acid measurements. Questionnaire for covariates: demographic, socioeconomic, and medical data (age, sex, smoking status, alcohol drinking status, education, and physical activity). Blood pressure. Fasting blood glucose.	Results may not be generalizable to other populations with different diet patterns, lifestyles, and genetic backgrounds; participants were young, so there was a low prevalence of major chronic disease.
Quebec Ice Storm (1998)	±6 h to 5 weeks	Natural	Negative impact on cognitive and language development of the unborn child, independent of maternal personality factors [59]. The level of prenatal maternal stress is associated with higher BMI levels and adiposity in children of ages 5.5, 8.5, 13.5, and 15.5 years [60]. DNA methylation mediates the impact of exposure to prenatal maternal stress on BMI and central adiposity in children at age 13.5 years [61].	I. *n* = 150 [59] II. *n* = 386 [60] III. *n* = 66 [61]		I. Lacking data on the immediate biological processes. II. No control group exists; sample was skewed to the middle-upper and upper classes and relatively small. III. Small sample size.
9/11 Terrorist attacks (2001)	±1 day	Manmade	Offspring at one year of age of mothers exposed to attacks on the World Trade Center during pregnancy show lower salivary cortisol levels [84].	*n* = 38	Probable PTSD and PTSD severity by PTSD checklist. Severity of depression assessment with Beck Depression Index. Demographic and medical information, exposure and pregnancy outcome. Salivary sample collection + determination free cortisol levels. Relationship maternal PTSD—cortisol and infant cortisol levels, impact of pregnancy trimester of exposure.	Small sample size. Maternal age, ethnicity, BMI, hours of sleep, wakefulness, and breastfeeding.
Hurricane Katrina, Louisiana (2005)	±1 day	Natural	No association seen between maternal stress and child temperament [63].	*n* = 288	[63] Interview 8–10 weeks postpartum. Temperament interview 1 year postpartum. Delivery questionnaire (basic demographic information, hurricane stress exposure, and social support). Interview information on living conditions before the storm, at evacuation and currently. PTSD Checklist—civilian-version screening tool. Edinburgh Postnatal Depression Scale. Symptoms Cheklist-090-Revised.	The evaluation of the infants’ temperament was performed by the mother, the means of the EITQ were consistently lower than the means provided by the authors, and study limited to English-speaking women.
Iowa Flood (2008)	±3 weeks	Natural	Greater experienced prenatal maternal stress predicted higher BMI at 30 months in offspring [69].	*n* = 103		Small sample size; sample consisted of primarily Caucasian women of relatively high SES; and no control group.
The QF2011 Queensland Flood, Brisbane (2011)	±36 h	Natural	Increased 4-year child anxiety symptoms. [36] Association with prenatal maternal stress and alterations in placental glucocorticoid system [37].	I. *n* = 181 [36] II. *n* = 230 [37]		I. Small sample size. II. Small sample size, no placentas of women not exposed to the flood, and no placentas of women exposed in the third trimester.
Superstorm Sandy (2012)	±1 day	Natural	Prenatal maternal depression amplified by Sandy; association with lower emotion regulation and greater distress at 6 months in neonates [66,67].	I. *n* = 408 [66] II. *n* = 318 [67]		I. Infant temperament was based on maternal report; maternal mood may introduce a potential bias and potential unknown confounders; the statistical design of our study does not allow for an analysis of the impact of the exact timing of exposure to Sandy in relation to the trimester of exposure. II. Temperament was measured solely via maternal report. Trimester-specific effects of Sandy on temperament were not examined.
Tailings dam breakage, Brazil (2015)	± hours [breakage itself]	Manmade	Increased premature birth [85].	*n* = 914,795		
COVID-19 pandemic (2019)	Years	Natural and manmade	?			

## 5. Conclusions

The COVID-19 disaster has hit swiftly and globally, and the pandemic and its aftermath are expected to be long-lasting. Medical and financial consequences of the pandemic are already acknowledged. However, preventive protocols anticipating the psychological effects for (future) parents and their offspring need to be implemented, since stress related to the COVID-19 effect will impact the next generation long after the pandemic itself has been resolved. The reported data in Table 2 of the effects of in utero exposure to disasters and associated maternal stress can be used as a guidance to set up regional, national, and international maternal and family support programs to intervene in the intergenerational transmission of stress. Follow-up programs of these mothers and children, during the first 1000 days and beyond, will enable us to pinpoint which time spans during the periconception period, pregnancy, and postpartum period are vulnerable windows of exposure to disasters and maternal stress. It will also allow for the timely detection and treatment of cognitive, behavioral, or physical effects and offers opportunities to evaluate the effects of treatment on both women and their offspring. The importance of timely detection and treatment is even more emphasized by the first reports that have made their entry, indicating that higher maternal fear of COVID-19 during pregnancy was associated with a 192 g reduction in infant birthweight [104] and that greater COVID-19-related prenatal stress was significantly associated with higher infants’ SLC6A4 methylation in seven CpG sites, which, in turn, was negatively associated with infants’ temperament, in particular with the infants’ positive affect at 3 months [131].

An important unique characteristic of the COVID-19 pandemic in our current society is the repetitive periods of imposed social isolation. The impact of stress is determined by the balance of the impact of the stressor and resilience. Resilience is fostered by supportive relationships, both by supporting a generally resilient stress-system and by supporting buffering of acute stress. Social isolation, and especially disruption of a social bond between a significant pair, in itself activates the HPA-system [132]. Hence, in the case of the COVID-19 pandemic, not only stress is increased, but resilience is decreased. The supportive relationships that normally support the buffering of stress [133,134,135] have now become a source of stress themselves, as they are disrupted by the COVID-19 mitigation measures [136]. Furthermore, the stress resulting from interpersonal stress and increased exposure to domestic violence [103,137], which is increased during the COVID pandemic, is especially detrimental to fetal development, as is reflected in increased risk of preterm birth and low birth weight [138]. The COVID-19 pandemic represents a disaster in many unprecedented ways, of which stress in general is a main concern. Intervention programs for women early in pregnancy aimed at stress-reduction have shown to be able to reduce perceived stress [139,140], and pregnant women at risk for experiencing significant levels of stress seem open to these interventions [141]. Whether a reduction in stress during pregnancy due to intervention programs will also lead to a decrease in prenatal-stress-related effects in the next generation is unknown. Preliminary research evaluating the effect of mindfulness and Cognitive Behavioral Therapy on offspring brain development shows promising results [142,143].

Attention has been paid to positive effects of the COVID-19 pandemic and its mitigation measures, such as a reduction of air pollution and preterm birth [29,30]. Unfortunately, many negative results, such as increased mental health problems, decreased access to healthcare, increased domestic violence, increased poverty and job-insecurity effects, will have an impact on the current generation, as well as future ones [12,144,145,146]. Importantly, these effects seem to impact populations with a lower socioeconomic status even harder, worsening their current and future health and socioeconomic inequalities [147,148,149,150,151]. Therefore, there is a need to develop preventive and supportive programs for future pandemics. To maximize the interpretation of transgenerational effects of exposure to disaster in general, intensifying international collaboration is mandatory. Lessons learned from the past could globally be developed into effective population-wide preventive and intervention programs [152]. These programs should already be designed and aimed for the first 1000 days, as well as beyond that period, aiding both pregnant women and their offspring, both pre- and postnatally, against potentially COVID-19-pandemic-related health and socioeconomic consequences. We need to make sure that the ripples of COVID-19 do not evolve into unmanageable waves for the coming generation(s).

## Figures and Tables

**Figure 1 ijerph-19-04710-f001:**
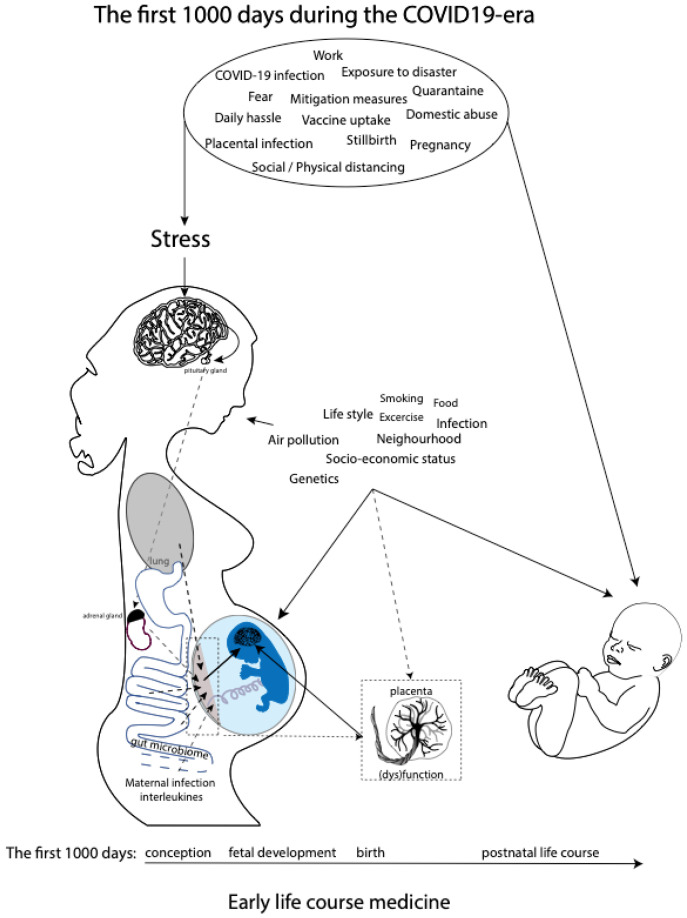
Overview of potential maternal prenatal stressors during the current COVID-19 pandemic as part of the early life course medicine. Several pathways affecting fetal (neuro)development in utero, including maternal HPA axis, maternal inflammation, maternal gut microbiome and placental dysfunction, and postnatal neurodevelopment during the first 1000 days, are shown.

**Table 1 ijerph-19-04710-t001:** Classification of disasters.

Type of Disaster	Subtypes	Examples
**Natural**	Biological	Viral pan-/epidemic
	Geological	Earthquake, vulcanic eruptions, tsunamis
	Hydrometerological	Floods versus droughts, storms
**Manmade**	Societal, hazardous, transportation, environmental	Terrorism, war, wildfires, Chernobyl meltdown, dam failure, ship wreckage, oil spillage

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
