# Peer review of "The Impact of Maternal Prenatal Stress Related to the COVID-19 Pandemic during the First 1000 Days: A Historical Perspective"

_ijerph, 2022, doi:10.3390/ijerph19084710_

Round 1

Reviewer 1 Report

Introduction:

  • Lines 66-71: here the authors list recommendations for expecting parents, but it is not mentioned in which country were these recommendations issued, or if this is a comprehensive list from multiple countries, each of which implemented only a part of the given list (if so, then I suggest changing the “and” to “or”). References are missing at the end of the sentence.
  • Line 77: vaccines, not vaccins
  • Lines 84-87: while it is true that maternal stress during pregnancy is associated with autism spectrum disorders and ADHD, the citations refer to studies that did not investigate that. Please cite appropriately. E.g. https://doi.org/10.3389/fpsyg.2010.00223, https://www.ncbi.nlm.nih.gov/pmc/articles/PMC6369590/
  • Line 89: reference missing.

Materials and methods:

  • Line 117: please explain what “a representative selection” is: what inclusion/exclusion criteria were used, and why.

Results:

  • Lines 125-126: the in utero exposure vs non-exposure means that the mothers were themselves sick with the Spanish flu, while the others were not, or are the groups from different times, one before/after and one during the 1918 influenza pandemic?
  • Line 168: at the first use of the term, please use the non-abbreviated form (HPA-axis)
  • Lines 160/171: Superstorm Sandy or Hurricane Sandy? Please name it consistently. Line 171: Sandy, not Sand.
  • Line 179: predispose, not predisposed.
  • Lines 192-193: …”long term alteration in the functioning of the HPA-axis”: in Table 1 you mention the “placental glucocorticoid system” – what do you mean by that? Explain the measurement performed or how the outcome variable(s) was/were assessed.
  • Line 203: led, not lead
  • Lines 238-241: why is this paragraph in italics? Please rephrase the last sentence, it is unclear.

Discussion:

  • Line 249: their, not its
  • Line 270: too many “and” conjunctions, where does the enumeration end? “iatrogenic” not iatrogen. It would perhaps be clearer to say what it is (if it refers to medically induced preterm birth due to signs of fetal distress).
  • Line 272: please detail which mental disorders are meant here.
  • Line 280: the presence OF patners
  • Line 281: Furthermore, also the risks of… => remove the “also” (Furthermore is enough)
  • Line 296: “are associated” -> is associated.
  • Line 304: “likely puts exposed these offspring” -> cut “exposed”. “… at an increased biological risk to develop PTSD themselves” – why, is lower cortisol correlated with higher PTSD risk? To what trauma would the offspring develop PTSD? Perhaps add “, should they later in life be exposed to a traumatic event”.
  • Line 328: add “as well as” between “stressors,” and “maternal subjective mental health complaints”

Prenatal maternal stress and Developmental Origins of Health and Disease

  • Line 332-334: I suggest moving the sentence “followed by… outcome of the child (108, 109)” after the phrase “such as fetal neurogenesis, neural migration, and myelination (104)”. Otherwise, it sounds as if the fetal neurogenesis, neural migration and myelination are adverse neurodevelopmental outcomes.
  • Please introduce the DOHaD paradigm a little: who proposed it, when.
  • Line 341: “its end product cortisol” – cortisol is just one of many “end products” of the HPA-axis, please rephrase, perhaps “most important end product” or “most studied”.
  • Lines 358-359: a verb is missing: “since an association between general maternal anxiety and composition of microbial stool sample.”
  • Line 370: “compromised placental function, including the 11b-HSD2 enzyme” – what about this enzyme, is it also dysfunctional in this situation, or is it activated or inhibited?
  • Line 371: “compromised maternal immune system functioning” – perhaps change to “dysfunctional maternal immune system”.
  • Line 372: “and nutritional behaviors including sleep, and exercise” – since previously you listed dysfunctions, perhaps here it should also be “poor sleep and lack/insufficient of exercise”.
  • Line 372: “(epi)genetics”: why the parentheses?
  • Lines 374-375: “fetal development and compromised offspring development” – is only the offspring development compromised?

Conclusions:

  • Lines 405-407: “supportive relationships (…) have now become a source of stress themselves, as they are disrupted by the COVID-19 mitigation measures.” – please cite the studies that name this connection between the measures and the stressful supportive relationships. Maybe they are stressful also from the perspective of a higher COVID-19 infection risk, fear of transmission from the supportive circle to the pregnant woman and eventually to the baby.
  • Line 413: “…and high-risk pregnant women” – high risk for what? Women with a high risk pregnancy, or at high risk of depression?
  • Lines 426-427: A verb missing? “… and interpret transgenerational effects of exposure to disaster in general, international collaboration.” What was meant with international collaboration, as there is no verb that connects to it in this sentence?

General:

In the methods or in the results, please mention, in a summarised fashion, the statistical tests that these studies performed, and whether you selected only medium or strong effects, or whether you filtered studies based on the type of statistical tests in any way.

Please also mention that different studies assessed different outcome variables, please mention all outcomes (+ citations of the studies involved) and mention also how that outcome was measured – for example, subjective stress = outcome, a specific questionnaire = the variable. “Cognitive and language development” (e.g. line 165 and Table 1 - Quebec Ice Storm) is very general – was this assessed via a test, an interview, a scale, an imaging technique? And at what age? Same for “maternal stress”, “emotion regulation”, “distress at 6 months in neonates” – please shortly name the method of measurement.

Table 1: line with 9/11 terrorist attacks: “show lower cortisol levels” – this was salivary cortisol, please change to “show lower salivary cortisol levels”

Figure 1: Line 352: affecting, not effecting fetal neurodevelopment.

Author Response

REVIEWER 1

Introduction:

  • Lines 66-71: here the authors list recommendations for expecting parents, but it is not mentioned in which country were these recommendations issued, or if this is a comprehensive list from multiple countries, each of which implemented only a part of the given list (if so, then I suggest changing the “and” to “or”). References are missing at the end of the sentence.

Answer: We now clarify in which countries these recommendations were issued

and included references. The original guidelines are unfortunately not available anymore due to the many updates over the last two years of the COVID-19 pandemic. However, the now mentioned references discuss these original guidelines (see Introduction).

  • Line 77: vaccines, not vaccins

Answer: Changed accordingly.

  • Lines 84-87: while it is true that maternal stress during pregnancy is associated with autism spectrum disorders and ADHD, the citations refer to studies that did not investigate that. Please cite appropriately. E.g. https://doi.org/10.3389/fpsyg.2010.00223, https://www.ncbi.nlm.nih.gov/pmc/articles/PMC6369590/

Answer: We apologies for the mistake. We have now included the appropriate references, including the 2 suggested ones.

  • Line 89: reference missing.

Answer: References are now included.

Materials and methods:

  • Line 117: please explain what “a representative selection” is: what inclusion/exclusion criteria were used, and why.

Answer: We have expanded our Materials and Methods section (see below) and added an overview table of types of disasters and several subtypes (see table 1).

Searches were performed in the databases of Medline, Embase, PubMed, Web of Science and Google Scholar between January 1900 until December 2021. The search strategy terms used included but were not limited to: prenatal stress, preconception period, transgenerational effects, natural disaster, man-made disaster, disaster, trauma, pandemic. These were combined using the Boolean operator ‘or’. As it is a narrative review, in- and exclusion criteria were not strictly set. However, only original articles in English language were included. To present an overview of effects of exposure to a disaster in general, a distinction was made into natural and man-made disasters (Table 1). As each disaster can differ between moment of occurrence, cause, intensity, extent or duration of exposure, a selection of the literature search was made based on review of titles and abstracts. A selection of published literature on effects of disaster, either natural or man-made, was made by SS and EJV based on: 1. time period of occurrence (i.e. aiming at different periods of the 20th and 21st century), 2. including examples of several subtypes of natural disasters and man-made disasters, 3. studies including follow up, or time between disaster and investigation of at least 5 years, and 4. including well-known disasters to appeal to imagination and recognition.

Results:

  • Lines 125-126: the in utero exposure vs non-exposure means that the mothers were themselves sick with the Spanish flu, while the others were not, or are the groups from different times, one before/after and one during the 1918 influenza pandemic?

Answer: Due to the height of the pandemic in the period Oct 1918 until Jan 1919, in utero exposure by maternal influenza infection was highly expected. Since the analyses are done on generalized data of the US population, actual infection cannot be verified. Therefore, the study concerns comparison between in utero groups during the height of the pandemic versus groups that were in utero beyond the period Oct 1918-Jan 1919. We clarified it as follows:

The 1918 influenza pandemic, also known as the Spanish Flu, started in the beginning of 1918 and ended after four waves in the beginning of 1920. In the USA, the 1918 pandemic influenza was highly concentrated from October 1918 until January 1919, and offspring potentially exposed in utero to maternal influenza infection were mostly born between January and September 2019. Several cohorts in the USA of offspring with potential in utero exposure to the 1918 influenza were compared to offspring that were in utero beyond the height of the 1918 pandemic. The results indicate significant adverse effects on several health and socioeconomic parameters across the life course for in utero exposure to 1918 influenza.

  • Line 168: at the first use of the term, please use the non-abbreviated form (HPA-axis)

Answer: Changed accordingly.

  • Lines 160/171: Superstorm Sandy or Hurricane Sandy? Please name it consistently.

Answer: We now use Superstorm Sandy throughout the manuscript.

  • Line 171: Sandy, not Sand.

Answer: Changed accordingly.

  • Line 179: predispose, not predisposed.

Answer: Changed accordingly.  

  • Lines 192-193: …”long term alteration in the functioning of the HPA-axis”: in Table 1 you mention the “placental glucocorticoid system” – what do you mean by that? Explain the measurement performed or how the outcome variable(s) was/were assessed.

Answer: We have added additional explanation to clarify the findings as follows:

Long term alteration in the functioning of the HPA-axis, including higher cortisol awakening responses, higher total daily cortisol secretion, and higher stress reactivity, evident from early ages onwards. 

In addition, based on questionnaires to assess maternal stress level, higher reported maternal stress was associated with an altered placental glucocorticoid system, as measured by a reduced presence of glucocorticoid receptor mRNA, which can be linked to reduced fetal protection against prenatal maternal cortisol (32-33).

  • Line 203: led, not lead

Answer: Changed accordingly.  

  • Lines 238-241: why is this paragraph in italics? Please rephrase the last sentence, it is unclear.

Answer: Our apologies for overlooking this error. The paragraph has been corrected and we deleted the last sentence.

Discussion:

  • Line 249: their, not its

Answer: Changed accordingly. 

  • Line 270: too many “and” conjunctions, where does the enumeration end? “iatrogenic” not iatrogen. It would perhaps be clearer to say what it is (if it refers to medically induced preterm birth due to signs of fetal distress).

Answer: We have changed the sentences into:

COVID-19 infection during pregnancy is also associated with adverse obstetrical outcomes, such as preeclampsia, (iatrogenic) preterm birth, and low birth-weight (120-122). In addition, pregnant women with COVID-19 are at risk for placental dysfunction, characterized by inflammatory changes, massive perivillous fibrin depositions and placental necrosis,  and related fetal distress, specifically when infected with the Delta variant (13-15).

  • Line 272: please detail which mental disorders are meant here.

Answer: We have added the following:

It is well-known that maternal infection in general during pregnancy poses a threat for development of a range of mental disorders in the offspring, such as schizophrenia, autism and attention deficit/hyperactivity disorder.

  • Line 280: the presence OF partners

Answer: Due to repetition of information in the introduction, and as suggested by another reviewer, we have removed a part of the paragraph, including this sentence.  

  • Line 281: Furthermore, also the risks of… => remove the “also” (Furthermore is enough)

Answer: Due to repetition of information in the introduction, and as suggested by another reviewer, we have removed a part of the paragraph, including this sentence.  

  • Line 296: “are associated” -> is associated.

Answer: Changed accordingly. 

  • Line 304: “likely puts exposed these offspring” -> cut “exposed”. “… at an increased biological risk to develop PTSD themselves” – why, is lower cortisol correlated with higher PTSD risk? To what trauma would the offspring develop PTSD? Perhaps add “, should they later in life be exposed to a traumatic event”.

Answer: We have changed the sentence into:

Since reduced levels of cortisol are linked to susceptibility to PTSD (87), these offspring are potentially at increased risk of developing PTSD when exposed to traumatic events themselves later in life (81).

  • Line 328: add “as well as” between “stressors,” and “maternal subjective mental health complaints”

Answer: Changed accordingly.

Prenatal maternal stress and Developmental Origins of Health and Disease

  • Line 332-334: I suggest moving the sentence “followed by… outcome of the child (108, 109)” after the phrase “such as fetal neurogenesis, neural migration, and myelination (104)”. Otherwise, it sounds as if the fetal neurogenesis, neural migration and myelination are adverse neurodevelopmental outcomes.

Answer: We have changed the sentence into:

The DOHaD paradigm (21) states that prenatal maternal stress can negatively affect pregnancy outcomes and fetal developmental processes, including neurogenesis, neural migration, and myelination (83). Adverse outcome of these fetal neurodevelopmental processes are associated with an increased risk of developing mental and physiological disorders later in life, such as autism spectrum disorder and attention-deficit/hyperactivity disorder (84, 85).

  • Please introduce the DOHaD paradigm a little: who proposed it, when.

Answer: We have added an introduction of the DOHaD paradigm in the introduction as follows:

This phenomenon is also described in the Developmental Origins of Health and Disease (DOHaD) paradigm (21), introduced by Barker and colleagues in 1986 (22). The DOHAD paradigm states that sensitive windows during fetal development exist, in which tissue development and function in utero and during the early postnatal life can be modified by external environmental exposures resulting in increased risks for adverse (neuro)developmental and health outcomes (23, 24).

  • Line 341: “its end product cortisol” – cortisol is just one of many “end products” of the HPA-axis, please rephrase, perhaps “most important end product” or “most studied”.

Answer: Changed as follows:

The maternal hypothalamic-pituitary-adrenal (HPA) axis and its most studied end product cortisol (…).

  • Lines 358-359: a verb is missing: “since an association between general maternal anxiety and composition of microbial stool sample.”

Answer: Changed as follows:

One of the mechanisms that have been proposed is the maternal gut microbiome (84), since an association seems to exist between maternal anxiety and composition of the maternal gut microbiota during pregnancy (90).

  • Line 370: “compromised placental function, including the 11b-HSD2 enzyme” – what about this enzyme, is it also dysfunctional in this situation, or is it activated or inhibited?

Answer: We have changed the sentence as follows:

compromised placental function (e.g. an inability to upregulate 11b-HSD2 expression necessary to reduce fetal exposure to cortisol (91, 92)…

  • Line 371: “compromised maternal immune system functioning” – perhaps change to “dysfunctional maternal immune system”.

Answer: Changed accordingly.

  • Line 372: “and nutritional behaviors including sleep, and exercise” – since previously you listed dysfunctions, perhaps here it should also be “poor sleep and lack/insufficient of exercise”.

Answer: Changed accordingly.

  • Line 372: “(epi)genetics”: why the parentheses?

Answer: The parentheses have been removed as suggested.

  • Lines 374-375: “fetal development and compromised offspring development” – is only the offspring development compromised?

Answer: We agree with the reviewer, and have changed the sentence into

prenatal maternal stress and compromised fetal and offspring development.

Conclusions:

  • Lines 405-407: “supportive relationships (…) have now become a source of stress themselves, as they are disrupted by the COVID-19 mitigation measures.” – please cite the studies that name this connection between the measures and the stressful supportive relationships. Maybe they are stressful also from the perspective of a higher COVID-19 infection risk, fear of transmission from the supportive circle to the pregnant woman and eventually to the baby.

Answer: We have added three recent studies investigating the moderating roles of social support and resilience on stress during the current COVID-19 pandemic as well as an article investigating the implications of the COVID-19 mitigation policies for national wellbeing.

  • Line 413: “…and high-risk pregnant women” – high risk for what? Women with a high risk pregnancy, or at high risk of depression?

Answer: We have changed the sentence into

pregnant women at risk for experiencing significant levels of stress seem open to these interventions

  • Lines 426-427: A verb missing? “… and interpret transgenerational effects of exposure to disaster in general, international collaboration.” What was meant with international collaboration, as there is no verb that connects to it in this sentence?

Answer: We have changed the sentence into:

To maximize interpretation of  transgenerational effects of exposure to disaster in general, intensifying international collaboration is mandatory.

General:

  • In the methods or in the results, please mention, in a summarised fashion, the statistical tests that these studies performed, and whether you selected only medium or strong effects, or whether you filtered studies based on the type of statistical tests in any way.

Answer: Since this manuscript involves a narrative review, and not a systemic review, we did not (re-)analyze, meta-analyze or filtered any of the included articles based on their statistical testing. As stated in the introduction, we aimed to present an overview of transgenerational effects due to prenatal maternal stress caused by maternal exposure to a diversity of disasters. We therefore feel that reporting statistical analyses and tests of the included studies are outside of the scope of the current narrative review. We did, however, expand our table with the sample size and the possible confounders and limitations per included study, to increase insight of the study designs used, and we included references (see Table 2). Lastly, we have changed the materials and methods into:

Searches were performed in the databases of Medline, Embase, PubMed, Web of Science and Google Scholar between January 1900 until December 2021. The search strategy terms used included but were not limited to: prenatal stress, preconception period, transgenerational effects, natural disaster, man-made disaster, disaster, trauma, pandemic. These were combined using the Boolean operator ‘or’. As it is a narrative review, in- and exclusion criteria were not strictly set. However, only original articles in English language were included. To present an overview of effects of exposure to a disaster in general, a distinction was made into natural and man-made disasters (Table 1). As each disaster can differ between moment of occurrence, cause, intensity, extent or duration of exposure, a selection of the literature search was made based on review of titles and abstracts. A selection of published literature on effects of disaster, either natural or man-made, was made by SS and EJV based on: 1. time period of occurrence (i.e. aiming at different periods of the 20th and 21st century), 2. including examples of several subtypes of natural disasters and man-made disasters, 3. studies including follow up, or time between disaster and investigation of at least 5 years, and 4. including well-known disasters to appeal to imagination and recognition.

  • Please also mention that different studies assessed different outcome variables, please mention all outcomes (+ citations of the studies involved) and mention also how that outcome was measured – for example, subjective stress = outcome, a specific questionnaire = the variable. “Cognitive and language development” (e.g. line 165 and Table 1 - Quebec Ice Storm) is very general – was this assessed via a test, an interview, a scale, an imaging technique? And at what age? Same for “maternal stress”, “emotion regulation”, “distress at 6 months in neonates” – please shortly name the method of measurement.

Answer: As we presented an overview of transgenerational effects due to prenatal maternal stress caused by maternal exposure to a diversity of disasters, and we aimed to maintain readability of the manuscript, we chose to not mention each different outcome variables in the text, but instead added an additional column describing the outcome variables per included study (see Table 2).

We have added three new columns to table 2:

  1. the sample size per included study
  2. outcome variables
  3. the confounders / limitations per included study

Table 1: line with 9/11 terrorist attacks: “show lower cortisol levels” – this was salivary cortisol, please change to “show lower salivary cortisol levels”

 Answer: Changed accordingly.

Figure 1: Line 352: affecting, not effecting fetal neurodevelopment.

Answer: Changed accordingly.

Reviewer 2 Report

General Feedback and Recommendations

  • Overall, the work described in this review manuscript is timely and important and provides a review of the effects of prenatal stress during COVID with historical context for potential short and long-term effects.
  • This manuscript will be relevant and interesting to IJERPH readership but needs some critical reorganization and expansion before it can be accepted.
  • The materials and methods really need to be expanded to more fully describe inclusion and exclusion criteria; how many articles were selected; was there any scale/criteria/threshold for how or why certain disasters/stress events were included. Types of stress markers/tests could be included here as measures as well as maternal vs fetal effects that were examined.
  • I feel that the content of the manuscript could be much more impactful and clear with some extensive reorganization. For example, there was excellent discussion of mechanisms and relationships between stress and effects on the fetus/child in the discussion section, but this would have been more helpful to include in the introduction and potentially incorporate at least some mechanistic interpretation into what you have laid out as your results section just to orient the reader. This could be a good space to introduce the model or other Life course theory/Fetal origins of disease models to help set the stage for your results/discussion. As it stands the discussion sections feels disconnected from the results and really does not focus on how (or why the focus on) type of disaster may be impacting this. Additionally, I think that organizing by disaster type in your results may be confusing- rather it might be helpful to focus on duration of disaster and/or studies that collected similar data for stress markers- this could help the flow from results to discussion.
  • The results section is lacking some detail. The table could be greatly expanded by a few columns- I think it would be helpful to the reader to see sample sizes included for the cited studies and if confounding factors were controlled or at least noted.
  • There are minor formatting issues. For example on page 2, line 54, you have two different in-text citation styles and missing references line 71, 89.
    • On page 5 lines 237 – 241 – this paragraph is italicized and does not seem to fit the content where it is placed. I am unsure if this was just an error or intentional.
  • There are a few minor grammatical/sentence structure issues. See lines 89-92 – this particular sentence is a bit unclear and lengthy as written.
  • There are no limitations mentioned in the conclusions especially in light of this being based on historical context where data collections procedures may have differed and things like resource allocation, communication, medical responsiveness, etc,. all could be potential confounders.

Discussion:

  • Page 6, lines 290- 293- The sentence stating that “greater COVID-19 related prenatal stress was significantly associated with higher infants’ SLC6A4 methylation in seven CpG sites, which in turn predicted infants’ temperament at 3 months (96).” reads as ambiguous and needs more explanation- were these temperaments negative and by what scale or comparison?

Author Response

REVIEWER 2:

General Feedback and Recommendations

  • Overall, the work described in this review manuscript is timely and important and provides a review of the effects of prenatal stress during COVID with historical context for potential short and long-term effects.

Answer: Thank you.

  • This manuscript will be relevant and interesting to IJERPH readership but needs some critical reorganization and expansion before it can be accepted.

Answer: We drastically reorganized the manuscript and our own progressive insight. We like to refer to the revised version, especially to the Discussion and other paragraphs marked in yellow.

  • The materials and methods really need to be expanded to more fully describe inclusion and exclusion criteria; how many articles were selected; was there any scale/criteria/threshold for how or why certain disasters/stress events were included. Types of stress markers/tests could be included here as measures as well as maternal vs fetal effects that were examined.

Answer: We have expanded our Materials and Methods section (see above and the manuscript) and added an overview table of (sub)types of disasters.

Table 1. Classification of disasters

Type of disaster

Subtypes

Examples

Natural

Biological

Viral pan-/epidemic

Geological

Earthquake, vulcanic eruptions, tsunami’s

Hydrometerological

Floods versus droughts, storms

Man made

Societal, hazardous, transportation, environmental

Terrorism, war, wildfires, Chernobyl meltdown, dam failure, ship wreckage, oil spillage

  • I feel that the content of the manuscript could be much more impactful and clear with some extensive reorganization. For example, there was excellent discussion of mechanisms and relationships between stress and effects on the fetus/child in the discussion section, but this would have been more helpful to include in the introduction and potentially incorporate at least some mechanistic interpretation into what you have laid out as your results section just to orient the reader. This could be a good space to introduce the model or other Life course theory/Fetal origins of disease models to help set the stage for your results/discussion. As it stands the discussion sections feels disconnected from the results and really does not focus on how (or why the focus on) type of disaster may be impacting this. Additionally, I think that organizing by disaster type in your results may be confusing- rather it might be helpful to focus on duration of disaster and/or studies that collected similar data for stress markers- this could help the flow from results to discussion.

Answer: We extensively reorganized the manuscript. First, to orient the reader, we now introduce the concept of fetal programming, underlying mechanisms and Figure 1 in the Introduction section. To maintain the flow of the Introduction, we refer the reader in the introduction to a more detailed discussion of the potential underlying mechanisms relating maternal prenatal stress to child outcomes to the Discussion section. To address the remarks concerning the fact that it feels that the discussion feels disconnected from the results and does not focus on how the type of disaster may be impacting on this, as well as the organization of the results, we would like to point that a recent review (Tuovinen et al. Maternal antenatal stress and mental and behavioral disorders in their children. Journal of Affective Disorders, 2021; 278: 57-65) mentions that associations between maternal antenatal stress and mental and behavioral disorders are not gestational week-specific and do not differ by type of stress. With our selection of included studies (see adapted Materials and Methods), we wanted to argument  that prenatal maternal stress is the main causative factor, and that the type of disaster or duration of exposure, although relevant, seems of minor importance. The COVID-19 pandemic is a disaster with at least an extensive duration and repetitive waves, which can all cause prenatal maternal stress.

  • The results section is lacking some detail. The table could be greatly expanded by a few columns- I think it would be helpful to the reader to see sample sizes included for the cited studies and if confounding factors were controlled or at least noted.

Answer: We have added three columns to the table:

  1. the sample size per included study
  2. outcome variables per included study
  3. the confounders / limitations per included study

  • There are minor formatting issues. For example on page 2, line 54, you have two different in-text citation styles and missing references line 71, 89.

Answer: We apologies for the mistakes, and removed 1 in-text reference, and added the missing references.

  • On page 5 lines 237 – 241 – this paragraph is italicized and does not seem to fit the content where it is placed. I am unsure if this was just an error or intentional.

Answer: Our apologies for overlooking this error. The paragraph has been corrected and we deleted the last sentence.

  • There are a few minor grammatical/sentence structure issues. See lines 89-92 – this particular sentence is a bit unclear and lengthy as written.

Answer: As the lines unfortunately do not seem to correspond anymore to the original lines mentioned by the reviewer, we believe the reviewer refers to the following sentence:

Therefore, historical experiences and lessons learned should be applied to prevent or timely detect similar consequences following the current COVID-19 pandemic, which without question qualifies as a disaster.

Which we have changed into:

We believe that the transgenerational lessons learned from historical experiences with disasters should quickly be applied to the current COVID-19 pandemic, to prevent or timely detect similar long term consequences.

  • There are no limitations mentioned in the conclusions especially in light of this being based on historical context where data collections procedures may have differed and things like resource allocation, communication, medical responsiveness, etc,. all could be potential confounders.

Answer: We now mention limitations in Table 2 and in the Discussion section.

Discussion:

  • Page 6, lines 290- 293- The sentence stating that “greater COVID-19 related prenatal stress was significantly associated with higher infants’ SLC6A4 methylation in seven CpG sites, which in turn predicted infants’ temperament at 3 months (96).” reads as ambiguous and needs more explanation- were these temperaments negative and by what scale or comparison?

Answer: We have clarified this sentence accordingly:

which in turn was negatively associated with infants’ temperament, in particular with the infants’ positive affect at 3 months (126).

Reviewer 3 Report

Thank you very much for the opportunity for me to read and review your manuscript. It addresses an emerging relevant public health issue of maternal prenatal stress related to the COVID-19 pandemic. It is overall well-written and I have included my comments below to strengthen the overall quality of the manuscript.

Author Response

REVIEWER 3

Thank you very much for the opportunity for me to read and review your manuscript. It addresses an emerging relevant public health issue of maternal prenatal stress related to the COVID-19 pandemic. It is overall well-written and I have included my comments below to strengthen the overall quality of the manuscript.

Answer: Thank you.

Abstract:

Line 31: What is the overall definition of reproductive population? Women 15-49 years old?

Answer: We have changed the sentence and term into  

women of reproductive age (15-49 year) …

Introduction:

Line 42: Should be written as “sensitive period for development of symptoms of anxiety, depression, and stress

Answer: Changed accordingly.

Lines 48-49: Should be written as “uncertainty about their partner’s”

Answer: Changed accordingly.

Line 75: “resulting in placental dysfunctions” Specifically, what do the authors mean by “placental dysfunctions”

Answer: We have changed the sentence into:  

Also, SARS-CoV-2 infection during pregnancy can lead to placental infection resulting in massive placental necrosis (13), with is related to fetal distress and increased risk for stillbirth (14, 15).

Line 77: Word “vaccins” is misspelled. Should be replaced with term “vaccines”

Answer: Changed accordingly.

Lines 80-81: “Pregnant women were in initial phases recommended against vaccination” Does the term “initial phases” refer to the first trimester of pregnancy? This should be clarified.

Answer: To clarify the section of the paragraph, we have changed it into:

Vaccine uptake during pregnancy is in general confronted with a variety of barriers (18). Due to exclusion of pregnant women from initial vaccination clinical trails (16), evidence of safety and efficacy of COVID-19 vaccination during pregnancy was lacking. Therefore, with the introduction of the COVID-19 vaccines, many countries were not offering COVID-19 vaccines to pregnant women (19), which changed after the Centers for Disease Control and Prevention report in December 2020 (17, 18).

Line 98: Replace term “overview” with “review”

Answer: Changed accordingly.

Line 104: Replace term “manuscript” with “narrative review”

Answer: Changed accordingly.

Materials and methods:

Lines 133-119: Were MeSH-compliant keywords used? What were the date ranges that the keywords were searched? Did the authors also include letters to the editor? On what basis were articles chosen for inclusion? What were the criteria used to select the articles?

Answer: We have expanded our Materials and Methods section (see above and the manuscript) and added an overview table of (sub)types of disasters (see Table 1).

Results:

Line 137: Unclear what the authors mean by "mid gestation". Is it the second trimester?

Answer: Yes, that is correct. We have changed mid gestation into second trimester.

Line 171: Replace term “Hurricane Sand” with “Hurricane Sandy”

Answer: Due to comments of another reviewer, we have change Hurricane into Superstorm, and changed Sand into Sandy.

Lines 194-195: Unclear what is meant by “infant outcomes”?

Answer: We have clarified infant outcomes, and have changed the sentence into

Moreover, greater pregnant women’s flood-related objective and subjective stress predicted worse and delayed developmental outcomes for fine motor and problem solving skills at six months of infant age (40).

Line 221: I would recommend adding “in Biafra, a former secessionist state in West Africa”

Answer: Changed accordingly.

Lines 233-234: I would replace “a significant shorter time of pregnancy” with “a shortened length of gestation”

Answer: Changed accordingly.

Discussion:

Line 249: I would replace “independent of its cause” with “independent of their cause”

Answer: Changed accordingly.

Line 268: Should be “obstetrical outcomes such as”

Answer: Changed accordingly.

Lines 269-270: Unclear what is meant by “placental dysfunction”

Answer: We have clarified placental dysfunction, by changing the sentence into:

In addition, pregnant women with COVID-19 are at risk for placental dysfunction, characterized by inflammatory changes, massive perivillous fibrin depositions and placental necrosis, and related fetal distress, specifically when infected with the Delta variant (13-15).

Line 270: Unclear what is meant by term “iatrogen”

Answer: We have changed the sentences into:

COVID-19 infection during pregnancy is also associated with adverse obstetrical outcomes, such as preeclampsia, (iatrogenic) preterm birth, and low birth-weight (120-122).

Line 271: Is the COVID-19 omicron variant also associated with placental insufficiency?

Answer: Until now, there no reports of placental infection by the omicron variant.

Line 272: Is there a specific trimester during pregnancy associated with development of mental disorders?

Answer: A recent review (Tuovinen et al. Maternal antenatal stress and mental and behavioral disorders in their children. Journal of Affective Disorders, 2021; 278: 57-65) mentions that associations between maternal antenatal stress and mental and behavioral disorders are not gestational week-specific and do not differ by type of stress.

Lines 273-274: Rephrase to “caused by an invading pathogen”

Answer: Changed accordingly.

Line 280: Should be “presence of partners”

Answer: Changed accordingly and moved the sentence up to the introduction:

Several professional associations in different countries, including The Royal College of Obstetrics and Gynaecology (United Kingdom), American College of Obstetrics and Gynaecology (USA) and the Dutch Society of Obstetrics and Gynaecology (the Netherlands), quickly responded with  recommendations for expecting parents such as minimizing hospital visits and restricting outpatient clinic consultations, remaining indoors and self-isolate when symptomatic for at least 7 days, delay routine obstetrical care until after the self-isolation period, physical distancing between physician and patient, restricting hospital deliveries, and restrictions around the presence of partners and significant others during labor (10-12).

Line 296: Should be “is associated with”

Answer: Changed accordingly.

Line 296: Specifically, what birth outcomes? Low birth weight? Preterm birth? Infant mortality?

Answer: We have changed the sentence as follows:

Prenatal exposure to negative maternal psychological factors, including stress due to disasters, is thus associated with a range of poor outcomes for the offspring, such as preterm birth, risk for schizophrenia later in life, and a dysbiotic gut microbiome, with some effects even lasting until the end of the life course (83, 86, 88, 95-118)

Line 304: Should be: “puts exposed offspring at an increased risk of developing PTSD themselves”

Answer: Also due to comments of other reviewers, we have changed the sentence into:

Since reduced levels of cortisol are linked to susceptibility to PTSD (87), these offspring are potentially at increased risk of developing PTSD when exposed to traumatic events themselves later in life (81).

Line 304: Delete term “biological”

Answer: Changed accordingly.

Line 304: Replace term “to develop” with “of developing”

Answer: Changed accordingly.

Line 313: “or even (a combination of) other, factor(s)”. What other factors? Could the authors provide some examples?

Answer: We agree with the reviewer that this was confusing, and have removed that specific part of the sentence.

Line 335: “increased risk of developing mental and physiological disorders”. Specifically, what types of mental and physiological disorders?

Answer: We have clarified this by adding:

Adverse outcome of these fetal neurodevelopmental processes are associated with an increased risk of developing mental and physiological disorders later in life, such as autism spectrum disorder and attention-deficit/hyperactivity disorder (84, 85).

Line 350: Unclear what is meant by “maternal cortisol markers”? Do the authors mean “maternal cortisol level”?

Answer: We have clarified this by adding:

maternal biomarkers of HPA-axis functioning, such as cortisol reactivity or the cortisol output during the day (25).

Line 408: “increased domestic violence”. Should be “increased exposure to domestic violence”

Answer: Changed accordingly.

Lines 409-410: “increased preterm birth and low birth weight”. Should be “increased risk of preterm birth and low birth weight”

Answer: Changed accordingly.

Conclusions:

Line 413: Unclear what the authors mean by “high-risk pregnant women”. Women exposed to stress? Women exposed to COVID-19?

Answer: We apologies for the confusion, and have clarified the meaning of high-risk pregnant women, by changing the sentence into:

…and pregnant women at risk for experiencing significant levels of stress seem open to these interventions (136).

Lines 418-419: Replace term “mitigation rules” with “mitigation measures

Answer: Changed accordingly.

Round 2

Reviewer 2 Report

The revisions look great and really help to strengthen the manuscript. Thank you for taking the time to address each clearly and succinctly.

This manuscript is a resubmission of an earlier submission. The following is a list of the peer review reports and author responses from that submission.